# Therapeutic Nanobodies Targeting Cell Plasma Membrane Transport Proteins: A High-Risk/High-Gain Endeavor

**DOI:** 10.3390/biom11010063

**Published:** 2021-01-06

**Authors:** Raf Van Campenhout, Serge Muyldermans, Mathieu Vinken, Nick Devoogdt, Timo W.M. De Groof

**Affiliations:** 1Department of In Vitro Toxicology and Dermato-Cosmetology, Vrije Universiteit Brussel, Laarbeeklaan 103, 1090 Brussels, Belgium; raf.van.campenhout@vub.be (R.V.C.); mathieu.vinken@vub.be (M.V.); 2Laboratory of Cellular and Molecular Immunology, Vrije Universiteit Brussel, Pleinlaan 2, 1050 Brussels, Belgium; serge.muyldermans@vub.be; 3In Vivo Cellular and Molecular Imaging Laboratory, Vrije Universiteit Brussel, Laarbeeklaan 103, 1090 Brussels, Belgium; ndevoogd@vub.be

**Keywords:** nanobodies, cell plasma membrane transport proteins, drug target, therapy

## Abstract

Cell plasma membrane proteins are considered as gatekeepers of the cell and play a major role in regulating various processes. Transport proteins constitute a subclass of cell plasma membrane proteins enabling the exchange of molecules and ions between the extracellular environment and the cytosol. A plethora of human pathologies are associated with the altered expression or dysfunction of cell plasma membrane transport proteins, making them interesting therapeutic drug targets. However, the search for therapeutics is challenging, since many drug candidates targeting cell plasma membrane proteins fail in (pre)clinical testing due to inadequate selectivity, specificity, potency or stability. These latter characteristics are met by nanobodies, which potentially renders them eligible therapeutics targeting cell plasma membrane proteins. Therefore, a therapeutic nanobody-based strategy seems a valid approach to target and modulate the activity of cell plasma membrane transport proteins. This review paper focuses on methodologies to generate cell plasma membrane transport protein-targeting nanobodies, and the advantages and pitfalls while generating these small antibody-derivatives, and discusses several therapeutic nanobodies directed towards transmembrane proteins, including channels and pores, adenosine triphosphate-powered pumps and porters.

## 1. Introduction

The cell plasma membrane is an essential structure, as it shapes cyto-architecture and protects cellular integrity. Although the cell plasma membrane features an impermeable double phospholipid layer structure, the passage of ions and biomolecules, such as glucose, glutamate, adenosine triphosphate (ATP) and water-soluble vitamins, through this membrane is essential for maintaining cellular homeostasis. Such cross-membrane trafficking of biomolecules and ions between the cellular cytosol and the extracellular environment is facilitated and regulated by membrane transport proteins that are associated with the cell plasma membrane [1,2].

Cell plasma membrane proteins form a family of integral membrane proteins that can be grouped in several subclasses, denoted as channels and pores, ATP-powered pumps and porters [3]. By enabling the transport of substances across the cell plasma membrane, these transport proteins are key molecules for the establishment of physiological processes in all vital organs [4]. Moreover, cell plasma membrane transport proteins are also frequently involved in disease [2,4]. As such, misfolding, mutations, downregulation or the overexpression of cell plasma membrane transport proteins are implicated in pathological conditions [2,5,6,7]. Consequently, cell plasma membrane transport proteins have gained considerable interest as potential drug targets [2,5,8,9]. However, the clinical translation of small-molecule drugs targeting these transport proteins is mostly impeded by the lack of selectivity, specificity, potency or stability of these molecules [2,10,11]. Moreover, membrane transport proteins are known to be highly dynamic, resulting in different protein conformations, which can interfere with identification/screening of drugs targeting cell plasma membrane transport proteins. The use of conventional antibodies and antigen-binding fragments could overcome some of these issues [12,13,14,15,16]. However, the identification of cell membrane transport protein-targeting antibodies may be limited due to the large size of antibodies and their generally more concave paratope, making small and hidden cryptic epitopes of these proteins potentially unreachable [17,18,19,20]. Nanobodies are a plausible therapeutic alternative for targeting cell plasma membrane transport proteins.

In this review paper, the biology and classification of membrane proteins is described, with a special focus on cell plasma membrane transport proteins. Moreover, the characteristics of antibodies and a state-of-the-art overview of methods to generate nanobodies against these transport proteins is presented. Finally, the use of transport protein-targeting nanobodies as therapeutic agents for various diseases is discussed.

## 2. Classification of Cell Plasma Membrane Proteins: Focus on Multi-Pass Transmembrane Proteins

Cell plasma membrane proteins form a superfamily of proteins that are connected to the outer phospholipid bilayer. These proteins can be classified based on their structure, topology or their function. Based on the former, cell plasma membrane proteins can be generally categorized into three general classes, namely integral membrane proteins, peripheral membrane proteins and lipid-anchored proteins (Figure 1A) [21].

The class of integral membrane proteins can be further subdivided into two groups based on their interaction with the cell membrane, namely integral monotopic proteins and transmembrane proteins (Figure 1B) [22,23]. Integral monotopic proteins interact with the membrane from one side (i.e., inside or outside) and do not cross the cell plasma membrane. The monotopic proteins are underrepresented in the family of integral membrane proteins, and are performing catalyzing reactions on plasma membrane-resident substrates [24]. In contrast, transmembrane proteins span across the cell plasma membrane at least once. These proteins are made up of one or more hydrophobic transmembrane domain(s), soluble regions at the extracellular and/or intracellular N- and C-termini, and some signal and/or anchor sequences to direct the translocation and correct insertion of these proteins into the cell plasma membrane [22,23]. Based on their topology, the group of transmembrane proteins is classified into several subgroups (Figure 1C) [22,25]. Type I-IV transmembrane proteins are described as single-pass membrane proteins. Their classification is based on the location of the N- and C-terminal ends of the proteins across the cell plasma membrane and the presence of signal peptides and anchor sequences. The structural understanding of these single-pass transmembrane proteins has recently been described in more detail [26]. Other types of transmembrane proteins have a more complex structure, including β-barrel and multi-pass transmembrane proteins [22,27]. The latter feature a complex assembly of both hydrophobic and hydrophilic protein domains allowing multi-pass transmembrane proteins to span the cell plasma membrane more than once [22,23,27]. The structural subgrouping of these proteins is complicated, as structural determination via crystallization or cryogenic electron microscopy remains challenging due to their dynamic and fragile character so that they unfold rapidly upon extraction from their phospholipid bilayer environment [28,29,30]. However, these multi-pass transmembrane proteins can be divided into different classes based on their function, such as transport proteins and receptors (Figure 1D) [23,27,31,32].

## 3. Structure and Function of Cell Plasma Membrane Transport Proteins

Cell plasma membrane transport proteins participate in the transfer of substances through the cell plasma membrane, including glutamate, glucose, ATP and cyclic adenosine monophosphate as well as ions, such as calcium, potassium and sodium [1,2]. This activity is crucial to maintain cellular homeostasis and plays a key role in the regulation of physiological processes [1,2,4]. In view of the critical role of cell plasma membrane transport proteins, modulation of their basal performance is associated with multiple pathologies [2,4,6,7]. Targeting these transport proteins, which include channels and pores, ATP-powered pumps and porters (Figure 1E), would be of therapeutic interest [2,5,8,9].

### 3.1. Channels and Pores

Channels and pores allow the passage of small molecules and ions through a water-filled tunnel, which is simultaneously exposed to the intracellular and the extracellular environment [33,34,35]. While channels and pores share a similar structure, their conformation is different. While pores are always open, channels can adopt either an open or a closed conformation (Figure 1E) [34]. Among the many channels are those built up by connexin (Cx) and pannexin (Panx) proteins [7,8,36]. Members of the connexin and pannexin family consist of 4 transmembrane regions, 2 extracellular loops, 1 cytoplasmic loop and an intracellular C-terminus and N-terminus [37,38]. At present, more than 20 different connexin isoforms have been identified in humans [39]. Connexin proteins are widely expressed and are named according to their molecular weight [36]. The best-known family member is Cx43, which is the most abundantly expressed and has been studied extensively because of its role in a variety of cellular processes and diseases [7,8,36,40]. Connexins control cellular communication via the formation of connexin hemichannels, which consist of six connexin molecules, and gap junctions arising from the interaction of two hemichannels of adjacent cells [7,8,36]. Unlike connexins, only three pannexins have yet been identified, which are named Panx1, Panx2 and Panx3, following their order of discovery [8,36,41]. Pannexins do not form gap junctions but build up hexameric pannexin channels reminiscent of connexin hemichannels [42]. Connexin hemichannels and pannexin channels facilitate paracrine communication and play a central role in the induction and propagation of cell death and inflammation [37,43,44]. The pharmacological closing of channels provides therapeutic opportunities in a variety of diseases [8,37,43,44]. However, the current lack of specific connexin hemichannel and pannexin channel inhibitors hinders clinical exploration in this direction [8,43,44,45].

### 3.2. ATP-Powered Pumps

ATP-powered pumps translocate ions and small molecules against their concentration gradient across the cell plasma membrane by utilizing energy from ATP hydrolysis (Figure 1E) [2,46,47]. A well-studied ATP-powered pump is the hydrogen potassium ATPase. The hydrogen potassium ATPase is a heterodimeric protein that is composed of a multi-pass transmembrane subunit and a type II transmembrane subunit [48,49]. Via the hydrolysis of ATP, it facilitates the exchange of hydrogen and potassium ions across the cell plasma membrane [46,47,48,49]. By doing so, the hydrogen potassium ATPase is responsible for acid production in the stomach in order to activate digestion. Inactivation of this cell plasma membrane transport protein results in decreased stomach acidity. At present, several proton pump inhibitors, like omeprazole and pantoprazole, are available as drugs for treating gastroesophageal reflux and peptic ulcer disease [46,47,48,49].

### 3.3. Porters

Porters mediate the passage of small molecules and ions like glucose and sodium and chloride ions [50]. The binding of substrates induces a conformational change in the porter protein, allowing the movement of these substrates across the cell plasma membrane [50]. Porters can be further classified based upon the number of types of molecules they convey [50,51]. Whereas uniporters transport one type of molecule, symporters and antiporters are classified as cotransporters that organize the exchange of two different substrates (Figure 1E) [50,51]. The role of porters as drug targets can be demonstrated by the group of solute carriers (SLC) [52,53].

## 4. Antibodies and Nanobodies to Target Cell Plasma Membrane Transport Proteins

### 4.1. Conventional Antibodies

The human adaptive immune system generates antibodies to protect against the continuous threat of infection and toxic compounds [54]. Hence, antibodies are naturally occurring therapeutics that specifically recognize and eliminate antigens. These characteristics allow the usage of antibodies as research tools as well as for clinical diagnosis or therapeutic purposes [54]. Immunoglobulin-G (IgG) is the most abundant antibody isotype circulating in blood of mammals and its structure is highly conserved throughout different species [54]. This Y-shaped protein molecule consists of two identical heavy (H) and two identical light (L) chains (Figure 2A) [55,56].

The L chain comprises one variable (VL) and one constant (CL) domain, whereas the H chain is built up by one variable (VH) and three conserved or constant (CH1, CH2 and CH3) domains. The CH2 and CH3 domains of the two H chains form the crystallizable fragment (Fc), which binds various cell receptors, such as Fc receptors and immune molecules, to generate an adequate immune response [19,55,56]. Furthermore, the two arms of the Y-shaped IgG molecule are known as the antigen-binding fragments (Fabs), of which the extremity comprises the paired VL and VH domains, referred to as the variable fragment (Fv) that associates to antigens [19,55,56]. This process of antigen recognition is mainly mediated by the hypervariable antigen-binding loops or complementarity determining regions (CDRs). Three CDR loops in the VH and three CDR loops in the VL are clustering at one end of the Fv forming the paratope with a surface that is complementary to the epitope, the surface that is recognized on the antigen [19,55,56]. The three parts (i.e., two Fabs and one Fc) of the IgG molecule are linked via a flexible hinge region, located between the CH1 and CH2 domains, permitting independent movement among the three regions [19,55,56].

Antibodies associate with high affinity to their cognate target and are very specific to their antigen [20,55]. As antibodies can be raised against virtually any possible molecule, antibodies are already being exploited in the discovery of therapeutics targeting cell plasma membrane transport proteins [12]. In this regard, lifastuzumab–vedotin is an antibody–toxin conjugate, where the antibody part is directed towards SLC34A2, a sodium-dependent phosphate symporter (i.e., multi-pass transmembrane protein). This antibody-toxin is currently tested in clinical trials for the treatment of various cancer types [57,58]. However, the use of therapeutic antibodies has also some disadvantages, as the large-scale production of antibody-based therapeutics is an extremely expensive process [59]. In addition, membrane transport proteins feature small and often cryptic epitopes, which could be difficult to target by conventional antibodies due to their large size (i.e., approximately 150 kDa) [12]. The introduction of antibody-fragments can partially resolve these problems. Smaller antibody entities, in particular, Fabs (i.e., approximately 60 kDa) and single-chain Fvs (i.e., approximately 28 kDa), are interesting alternatives to intact antibodies [55,59,60]. Nevertheless, the identification of therapeutic antibody(-fragments) recognizing cell plasma membrane transport proteins remains challenging due to the large flat or concave paratope of conventional antibodies, which has limited access to cryptic and conserved sites on these targets [19,61,62].

### 4.2. Nanobodies

Camelidae, including *Camelus dromedarius*, *Camelus bactrianus*, *Lama glama*, *Lama guanicoe*, *Vicugna pacos* and *Vicugna vicugna* possess, as well as conventional heterotetrameric antibodies, unique heavy-chain-only antibodies (HCAbs) [63,64]. These HCAbs are smaller than conventional antibodies, as they are devoid of L chains and the CH1 domain is absent from their H chain (Figure 2B). The HCAbs from camelids recognize antigens by only one single variable domain, known as the variable domain of a H chain of HCAbs (VHH). The VHH fragment, also referred to as nanobody, can be produced recombinantly by a variety of host cells, including, bacteria, yeasts, plants and mammalian cells [18,19,20].

Although nanobodies are the smallest, functional, intact antigen-binding fragments, they are still able to selectively target epitopes selectively and with high affinity. Whereas conventional antibodies and their Fv fragments have a paratope consisting of six CDRs (i.e., three in a VH and three in a VL domain), nanobodies only have three CDRs [18,19,20]. Nanobodies are believed to have larger CDRs, more mutation hotspots and recombination signal sequence mimics to compensate for missing VH-VL combinatorial diversity [65,66,67]. Moreover, the smaller size of the footprint and the generally more convex paratope allow nanobodies to target cryptic epitopes, such as the substrate binding site of membrane transport proteins, which are less accessible for conventional antibodies and their derivatives such as the Fab [12,19,61]. Furthermore, the single-exon origin (i.e., approximately 360 nucleotides), the intrinsic low immunogenicity, facile blood vessel extravasation, good tissue penetration, robustness upon exposure to extreme conditions and tolerance towards engineering of nanobodies offer advantages for various in vitro and in vivo applications [18,19,20].

Therapeutic nanobodies targeting cell plasma membrane transport proteins are being developed to interfere with the function of these channels and pores, ATP-powered pumps and porters [2,5,8,9]. Such therapeutic nanobodies may exert these functional effects via different mechanisms. They could block channels and pores or influence ligand binding (i.e., acting as orthosteric or allosteric modulators) resulting in decreased or enhanced ligand binding [68,69,70]. Furthermore, nanobodies could exert their therapeutic effect by stabilizing a particular conformational state (i.e., active or inactive) of cell plasma membrane proteins [18]. However, finding these membrane transport protein-targeting nanobodies is difficult. While protocols to generate nanobodies against soluble proteins are well-established, the identification of nanobodies directed towards membrane proteins, such as membrane transport proteins, is more challenging [71].

### 4.3. Identification of Antigen-Specific Nanobodies

For the identification of antigen-specific nanobodies, it is important to start with high-quality libraries of nanobodies [20]. Gene banks that represent a large number of nanobodies with maximal diversity are envisaged for the retrieval of target-specific nanobodies. To achieve the latter, different types of libraries (i.e., immune, synthetic and naïve) can be used [20]. Both immune and naïve nanobody libraries are based on naturally occurring HCAbs isolated from the peripheral blood lymphocytes of camelids. Whereas immunized camelids are used for the generation of immune libraries, the blood of non-immunized camelids is taken to construct naïve libraries. Synthetic libraries, based on a single or few nanobody frameworks that are subjected to diversification of the amino acids located in the paratope, have emerged as an alternative to naïve and immune libraries in the last few years [20,72,73,74,75].

The employment of immune libraries is a well-established approach to identify a diversity of antigen-specific nanobodies with a high success rate [20,76]. Immunizing a camelid with soluble and properly folded proteins mixed with adjuvant is the first step to elicit an affinity matured immune response in the HCAb classes and to generate an immune library [20,76,77]. Following multiple subcutaneous injections of an immunogen, the mRNA extracted from blood lymphocytes of the immunized camelid serves as a template for the reverse transcription to produce cDNA. The nanobody cDNA is amplified by polymerase chain reaction and ligated in a phagemid vector. Finally, bacteria are transformed with the ligated material. In order to ensure high quality, libraries should have a size of around 10^7^–10^8^ individual transformants, of which more than 70% should carry a phagemid with a nanobody-inserted sequence [20,76,77]. To secure a very high (i.e., close to 100%) number of clones with a nanobody insert of the proper length, the use of the Golden Gate cloning strategy might be considered. The nanobody should hereby substitute a lethal ccdB gene in the phagemid to allow bacteria to grow [76]. The antigen-specific nanobodies are retrieved from such large libraries after their expression at the tip of phages and selection by biopanning. During biopanning, multiple strategies can be followed to select the nanobodies with the highest affinity and specificity against the target of interest [20,76,77]. Moreover, different enrichment approaches can be developed to select for nanobody characteristics, such as affinity, specificity, blocking of ligands or protein–protein interactions [77,78].

### 4.4. Identification of Cell Plasma Membrane Protein-Targeting Nanobodies

The identification of cell plasma membrane protein-binding nanobodies is not an easy task, as the availability of a pure and properly folded target protein is a major requirement for the immunization of camelids and subsequent panning [71,77]. The use of recombinant proteins from the extracellular domain of cell plasma membrane protein for both immunization and panning purposes forms an elegant solution to bypass the difficulty to obtain an intact cell plasma membrane protein [12,79]. However, this strategy is only practical for single-pass membrane proteins, as multi-pass transmembrane proteins mostly lack a large identifiable extracellular domain that can be produced in its native conformation to act as surrogate targets [12,79,80,81,82,83,84]. Nevertheless, alternative immunization and panning strategies have successfully been developed to generate nanobodies targeting both intracellular and extracellular epitopes of multi-pass transmembrane membrane proteins (Table 1).

#### 4.4.1. Transfected Cell Lines

One strategy to retrieve nanobodies against multi-pass transmembrane proteins involves the utilization of cells that are stably or transiently transfected to express the protein in their plasma membrane. The use of mammalian cell lines ensures the proper folding and native conformation of the cell plasma membrane target of interest [71]. The employment of transfected cells to immunize a camelid will also elicit an immune response against other (immunodominant) components expressed on the host cell surface, which might complicate subsequent selection of target-specific nanobodies [17,71]. To tackle this shortcoming, it is recommended to use dromedary-derived cells as a host cell for the transfection, since these cells will be less immunogenic in camels or llamas [17,85]. Obviously, stably transfected or transduced cells with a high surface expression level of the transgene will increase the success rate in finding target-specific nanobodies [17,71]. Besides immunizations, transfected or transduced cells can also be used in the subsequent panning procedures [84,86]. The use of cell-based pannings allows the identification of nanobodies targeting the extracellular side the cell plasma membrane protein under scrutiny [84]. To avoid the enrichment of nanobodies binding to antigenic components of the host cell, it is recommended to use different cellular backgrounds for the immunizations and panning [17,71]. Moreover, switching to different host cells in consecutive rounds of panning might reduce the retrieval of binders of host cell antigens [86]. In addition, performing a negative selection whereby the assembled library is incubated with a cell line lacking the expression of the protein of interest, prior to incubation with transfected or transduced cells, helps to remove unspecific binders [87,88].

#### 4.4.2. Membrane Extracts

Apart from intact cells, one could consider using the membrane extracts of (transfected) cells for immunization. Besides the solubilization of cell plasma membrane proteins with detergents, cells can also be disrupted to generate membrane vesicles or fragments exposing both extracellular and intracellular epitopes of the transgene [71]. It is clear that high expression levels of the antigen will lead to a better immunization [71]. Membrane extracts are a valid alternative to whole cells when one wants to obtain nanobodies targeting both extracellular and intracellular epitopes. Similar to whole cells, these membrane extracts can be used during panning rounds [71,89,90].

#### 4.4.3. Nanodiscs

Traditional cell plasma membrane models, like micelles, bicelles and liposomes, are a common source for expressing cell plasma membrane proteins [91]. However, these cell plasma membrane models face limitations imposed by the employed detergents, causing the structure and function of cell plasma membrane proteins to alter [92]. Hence, the generation of nanodiscs is gaining more attention to maintain membrane proteins in their native form. Preparation of this stable and monodisperse cell plasma membrane model includes osmotic lysis of cells expressing the protein of interest in the presence of phospholipids and membrane scaffold proteins [93]. In this way, nanodiscs are produced that display the target protein [93]. The use of nanodiscs as immunogen during immunizations and subsequent panning can identify both intracellular- and extracellular-binding nanobodies [94]. Moreover, employing nanodiscs as immunogen is advantageous over immunizations with intact cells and membrane extracts, since it does not bear the risk of retrieving unspecific nanobodies that target other membrane proteins that are present on cells [79,95,96]. Nevertheless, the application of nanodiscs warrants some optimization for each antigen as the amount of the target and scaffold protein should be balanced for the successful expression of the target of interest [93,95].

#### 4.4.4. Virus-Like Particles

Virus-like particles (VLPs) mimic viral structural proteins and can be designed to incorporate cell plasma membrane structures [97]. In addition, VLPs are not infectious as they lack essential genomic material [71,97]. The expression of membrane proteins on VLPs can be recombinantly fabricated in various production platforms [98]. In this light, mammalian cells, such as human embryonic kidney (HEK) cells, can produce structural viral polyproteins along with the protein of interest [71,98]. The viral proteins’ self-assemble and buds from the host cell to form non-infectious VLPs embedded with the target protein [97,98]. Despite the high costs associated with the production, purification and characterization of VLPs, these VLPs offer benefits for the development of nanobodies that target membrane proteins [71,99]. Since this technology can display the extracellular domains of membrane proteins in high concentrations and in a properly folded and stable way without expressing unrelated membrane proteins, VLPs are used to identify target-specific nanobodies. Thus, the applicability of VLPs is dual as the multiple subcutaneous injection of a camelid with VLPs raises an immune response towards the membrane protein of interest, and/or the panning on VLPs can be performed [71,99].

#### 4.4.5. cDNA Immunization

The immunization of a camelid with cDNA of the target cell plasma membrane protein in an expression vector forms an attractive alternative to the classical protein immunization [100]. The objective is to have the expression vector taken up by host cells, where transcription and translation will expose the membrane protein to the immune system to elicit an immune response against the protein of interest in its native form [84,100]. During the injection of the expression vector in the shoulders and hind limbs of camelids, an electroporation will introduce the DNA to host cells [100]. The main advantage of this strategy is that there is no need to purify the protein or to use cell plasma membrane models to present the antigen to the host animal immune system [100,101]. In this way, the risk of generating undesired binders is circumvented since the camelids’ immune system is only triggered by proteins encoded by the cloned target gene [100]. A disadvantage of this technique might be that the immune response of the host animal fails to elicit a strong immune response to the target [100]. However, this can be overcome by the inclusion of additional boosts with transfected cells expressing the target protein [68].

### 4.5. Therapeutic Nanobodies Targeting Cell Plasma Membrane Transport Proteins

#### 4.5.1. Channels and Pores

Nanobodies modulating channel activity are of considerable therapeutic relevance (Table 2). A case study in this respect includes nanobodies that block or potentiate gating of P2X7 channels [68,102]. Upon activation, trimeric P2X7 channels mediate the transport of calcium, sodium and potassium ions, thereby playing a central role in inflammatory diseases through inflammasome activation and the release of pro-inflammatory cytokines [103,104]. Llamas were immunized with HEK cells that stably express mouse or human P2X7 or cDNA encoding mouse and human P2X7 followed by a boost with mouse and human P2X7-transfected HEK cells. Subsequent biopanning on these transfected HEK cells gave rise to the identification of 18 different families of P2X7-targeting nanobodies [68]. Of these 18 families, six were able to block or enhance the activation of mouse P2X7 and two blocked ATP-mediated gating of human P2X7. Two of the mouse P2X7-targeting nanobodies were selected for further characterization, namely the antagonistic nanobody 13A7 and the agonistic nanobody 14D5. Both nanobodies were reformatted into bivalent formats and nanobody Fc-fusion proteins, which resulted in enhanced affinity and potency in the low nanomolar/sub-picomolar range [68]. Further in vitro characterization showed that bivalent formats of both nanobodies were able to modulate P2X7-induced ATP activation on primary mouse macrophages and T cells [68]. Next, the bivalent 13A7 nanobody was reformatted to a half-life extended 13A7 (13A7-HLE) nanobody via the addition of an anti-albumin nanobody to assess the therapeutic potential in vivo. The treatment of mice suffering from allergic contact dermatitis with 13A7-HLE resulted in reduced ear swelling and reduced levels of inflammatory cytokines [68]. Moreover, 13A7-HLE treatment in an antibody-induced glomerulonephritis model leads to a decrease in inflammatory cell infiltration and proteinuria in mice [68]. The clinical potential of these P2X7-binding nanobodies was further substantiated by Dano1, a human P2X7-specific nanobody. Reformatting of the Dano1 nanobody into a nanobody-Fc fusion resulted in enhanced potency and efficacy compared to the monovalent format and was able to lower the release of the pro-inflammatory cytokine IL-1β from endotoxin-exposed human monocytes [68,102].

Drugs that selectively target P2X4 channels equally offer therapeutic potential [105,106]. P2X4 channels mainly mediate the trafficking of calcium ions in response to ATP and are involved in different pathologies [105,107,108,109]. Given the lack of specific and potent P2X4-antagonists, studies have focused on the generation of nanobodies directed towards P2X4 [84]. Llamas were immunized with plasmids encoding either mouse or human P2X4 and P2X4-transfected HEK cells. Following biopanning on HEK cells expressing mouse or human P2X4, several nanobodies targeting extracellular regions of the P2X4 proteins could be retrieved. The specificity of these nanobodies was demonstrated by immunocytochemistry analysis of Chinese hamster ovary cells that were transfected with expression vectors encoding mouse or human P2X4 [84]. Furthermore, seven of the retrieved P2X7-targeting nanobodies were subcloned into bivalent formats to form a bivalent nanobody-rabbit IgG heavy chain antibody. The cross-reactivity of generated constructs was explored by flow cytometry analysis with HEK cells transfected to express mouse, rat or human P2X4. Two of the bivalent nanobody-rabbit Ig heavy-chain antibodies (Nb271-rbhcAb and Nb284-rbhcAb) recognized mouse, rat and human P2X4, whereas the other reformatted nanobodies showed affinity for only one or two of the P2X4 species [84]. Flow cytometry experiments confirmed the binding of endogenous murine P2X4 by Nb271 and Nb325. P2X4 expressed by mouse peritoneal mast cells and bone-marrow-derived macrophages were specifically targeted by the Nb271-rbhcAb [84]. However, the functional effects of identified P2X4-binding nanobodies have not been thoroughly investigated [84].

Another type of ion-channel-targeting nanobodies are Kv1.3-interacting nanobodies [110]. The Kv1.3 channel is a tetrameric structure, comprising multi-pass transmembrane proteins that mediate the voltage-dependent potassium ion permeability to control the activity of T effector memory cells [111,112]. Blocking these channels is a promising strategy for the treatment of chronic immune diseases like multiple sclerosis and type-1 *diabetes mellitus* [111,112]. Nanobodies targeting Kv1.3 channels inhibit the activity of human Kv1.3 channels in electrophysiological assays in a dose-dependent manner [110]. The formatting of these blocking nanobodies into bivalent formats improves the affinity of Kv1.3-targeting nanobodies from the low-nanomolar to the sub-nanomolar ratio and the construction of bi- and trivalent structures is beneficial for its functional activity as well [110]. This blocking effect of Kv1.3 binders results from recognizing a previously unidentified epitope in the first extracellular loop of these multi-pass transmembrane proteins [110]. Moreover, treatment of human T cells with a monovalent, a bivalent and a trivalent Kv1.3-binding nanobody resulted in T cell activation [110]. Finally, the therapeutic potential of the Kv1.3-targeting nanobodies was demonstrated by showing a reduction in the ear swelling response upon treatment with both half-life and non-half-life extended bivalent nanobodies in a hypersensitivity rat model [110].

#### 4.5.2. ATP-Powered Pumps

The zinc-transporting ATPase ZntA from the bacterium *Shigella sonnei* (SsZntA) is a transport protein from the P-type ATPase family [113]. Targeting and modulating pathogen-derived P-type ATPases is a promising strategy for the development of new antibiotics, antifungals, vaccines and herbicides, as these ATPases play important roles in the survival of pathogens (Table 2) [113]. The immunization of a llama with purified SsZntA and panning on biotinylated SsZntA resulted in the identification of multiple SsZnt-targeting nanobodies that could be grouped into three families [114]. One of the retrieved SsZntA-targeting nanobodies could selectively target the ZntA ATPase and significantly reduced its pump function by up to 50% [114]. However, the mechanism-of-action of this inhibition is not yet fully understood [114].

Another example of nanobodies targeting ATP-powered pumps includes the development of BtuCD-F-targeting nanobodies. The BtuCD-F transporter complex, built up by two multi-transmembrane proteins, participates in vitamin B12 import in bacteria [115,116]. Therefore, the identification of drugs that selectively block the uptake of essential nutrients forms the basis to design innovative antibiotics [115,116]. Nanobodies binding the periplasmic-binding protein BtuF were identified by injecting an alpaca with vitamin B12-bound BtuF and subsequent panning against vitamin B12-bound BtuF and apo-BtuF [116]. In total, six different BtuF-binding nanobodies were retrieved that could inhibit vitamin B12 binding to BtuF with inhibition constants ranging from sub-nanomolar to high nanomolar values [116]. Moreover, these BtuF-targeting nanobodies were able to partially inhibit the BtuCD-F-mediated substrate transport in *Escherichia coli*-derived spheroplasts [116]. Crystallization of the lead nanobody in complex with BtuF revealed that this nanobody exerted its effect by sterically hindering the vitamin-B12-binding pocket of BtuCD-F [116].

#### 4.5.3. Porters

Currently, nanobodies targeting porters are mainly used as crystallization chaperones [15]. A relevant case study in this respect relates to a SLC-26Dg-targeting nanobody [117]. SLC-26Dg acts as a symporter by combining the uptake of fumarate with the transport of protons in *Deinococcus geothermalis* bacteria [117]. Due to its high degree of homology with other SLC-26 proteins, the crystal structure of SLC-26Dg, obtained via the SLC-26Dg-targeting nanobody, has revealed valuable information on the structure and functional behavior of similar multi-transmembrane porters [117].

The identification of nanobodies targeting porters is also promising for the development of therapeutic agents (Table 2). Vesicular glutamate 1 (VGLUT1) is a multi-transmembrane porter belonging to the SLC17 family that loads glutamate into synaptic vesicles [118,119,120]. By doing so, VGLUT1 plays an important role in neurotransmission [119,120]. Moreover, changes in the activity or expression of this symporter is described in diseases, such as schizophrenia and epilepsy [119]. VGLUT1-targeting nanobodies were identified by immunizing a llama with a truncated and non-glycosylated mutant of the rat VGLUT1 protein. Subsequent biopanning on rat VGLUT1 mutant protein resulted in the identification of four nanobodies [121]. Despite being generated with a mutant protein, the anti-VGLUT1 nanobodies could still bind endogenous VGLUT1 expressed on mouse primary cortical neurons [121]. Further characterization of these nanobodies showed that the nanobodies recognized an intracellular, cytoplasmic epitope and were able to inhibit glutamate uptake in proteoliposomes and synaptic vesicles [121].

## 5. Conclusions and Future Perspectives

Cell plasma membrane proteins control biological processes, but equally underlie a wide spectrum of pathologies [2,4]. Compromised trafficking of molecules and ions across the cell plasma membrane may trigger disease or dysregulate signaling pathways towards pathology [4]. The development of drugs that restore cell plasma membrane transport protein function, therefore, is a promising avenue to explore [2,9]. However, small-molecule drug discovery in this research area is complicated since cell plasma membrane transport proteins share common structural properties, resulting in the non-specific binding of drugs targeting this class of proteins [2,10,11]. In this regard, aptamers or proteinaceous affinity reagents (DARPins, monobodies, affibodies, anticalins, or knob-like structures from cow antibodies) recognizing specifically membrane transport proteins can be an interesting alternative. In this review, we focused exclusively on nanobodies, as they are easily obtainable from immune libraries and free to use in therapeutic applications. All other formats or affinity reagents are only available from groups specialized in generating good quality, vast and diverse repertoires and handling these libraries and/or processing the techniques and skills for subsequent affinity maturation. The therapeutic use of such affinity reagents will always somehow be restricted.

Given their therapeutic potential, the identification of nanobodies modulating the activity of cell plasma membrane transport proteins is a relevant approach as this could offer new possibilities for the treatment of a variety of human diseases [18,19]. To date, only a limited number of examples has been described. However, multiple nanobodies targeting other classes of multi-pass transmembrane proteins, such as G-protein coupled receptors, have been demonstrated on many occasions, making it conceivable to assume that this would also be feasible for a broad diversity of cell-plasma membrane transport proteins.

At present, both monovalent and multivalent formats of cell plasma membrane transport protein-targeting nanobodies have been described. Interestingly, monovalent nanobodies can suffice to modulate the transfer of many substances and ions through the cell plasma membrane. By doing so, there is an interest in nanobodies that exert an agonistic or antagonistic effect [68]. Moreover, specific biopanning strategies and functional assays can be developed to retrieve agents that adapt influx and efflux processes mediated by cell plasma membrane transport proteins from nanobody libraries. Therapeutic nanobodies may here act as orthosteric or allosteric modulators and can overcome a lack of selectivity, specificity, potency or stability in small-molecule drugs by targeting small and hidden cryptic epitopes of these transport proteins [17,18,19,20,68,69,70]. Nevertheless, the reformatting of monovalent nanobodies to multivalent formats (i.e., bivalent, biparatopic or nanobody-Fc fusions) could be advantageous for therapeutic application [122,123,124,125,126]. It has been suggested that such agents are able to improve the envisaged functional effects based on an increase in avidity, potency and/or efficacy [18,127]. Another reason to reformat monovalent (non-modulating) nanobodies to multivalent would be the possibility of changing the functional characteristics of these nanobodies [127]. Besides changing the functional properties of the nanobodies, the reformatting of therapeutic nanobodies is also recommended for in vivo administration. To keep a constant level of circulating nanobodies in patients, it is necessary to compensate for the low molecular weight and small size of monovalent nanobodies [127,128,129]. PEGylation and PASylation of nanobodies, or their association with affinity reagents that target serum albumin or the Fc of IgG prolongs the serum half-life of antibody fragments [127,128,129]. Furthermore, nanobodies intended for therapeutic use can also be humanized, i.e., mutating camelid-specific amino acid sequences to their human equivalents, to reduce the possible risks of immunogenicity in patients [130].

Besides acting as antigen modulatory therapeutics, nanobodies can also be used as a vehicle to direct effector molecules towards a target antigen. At present, the application of nanobodies for targeted therapy has been mainly demonstrated for several classes of receptor proteins that are overexpressed on cancer cells [18]. In targeted therapy, nanobodies binding extracellular parts of overexpressed proteins are coupled with therapeutic agents via chemical conjugation or gene fusion [131]. In this way, nanobodies redirect molecules to specific organ systems, cell types or cell compartments [132]. Examples of molecules that have been fused to nanobodies include chemotherapeutics, toxins, radionuclides, photosensitizers, T cell and natural killer cell engagers [18,131,132]. Furthermore, nanobodies can be linked to liposomes, micelles or polymer particles containing hydrophilic or toxic drugs and redirect this payload to cells overexpressing the target receptor protein [18,131]. The specificity of nanobodies towards cell plasma membrane transport proteins might also be envisaged for these targeted therapy applications. In this light, nanobodies that bind cell plasma membrane transport proteins that are overexpressed in pathological conditions can deliver drugs that lack selectivity, specificity or stability towards the target cells of interest.

In summary, the generation of nanobodies targeting cell plasma membrane transport proteins remain a challenging endeavor due to the nature of these proteins. However, nanobodies targeting these proteins have significant therapeutic potential in a wide range of applications, which remains to be explored.

## Figures and Tables

**Figure 1 biomolecules-11-00063-f001:**
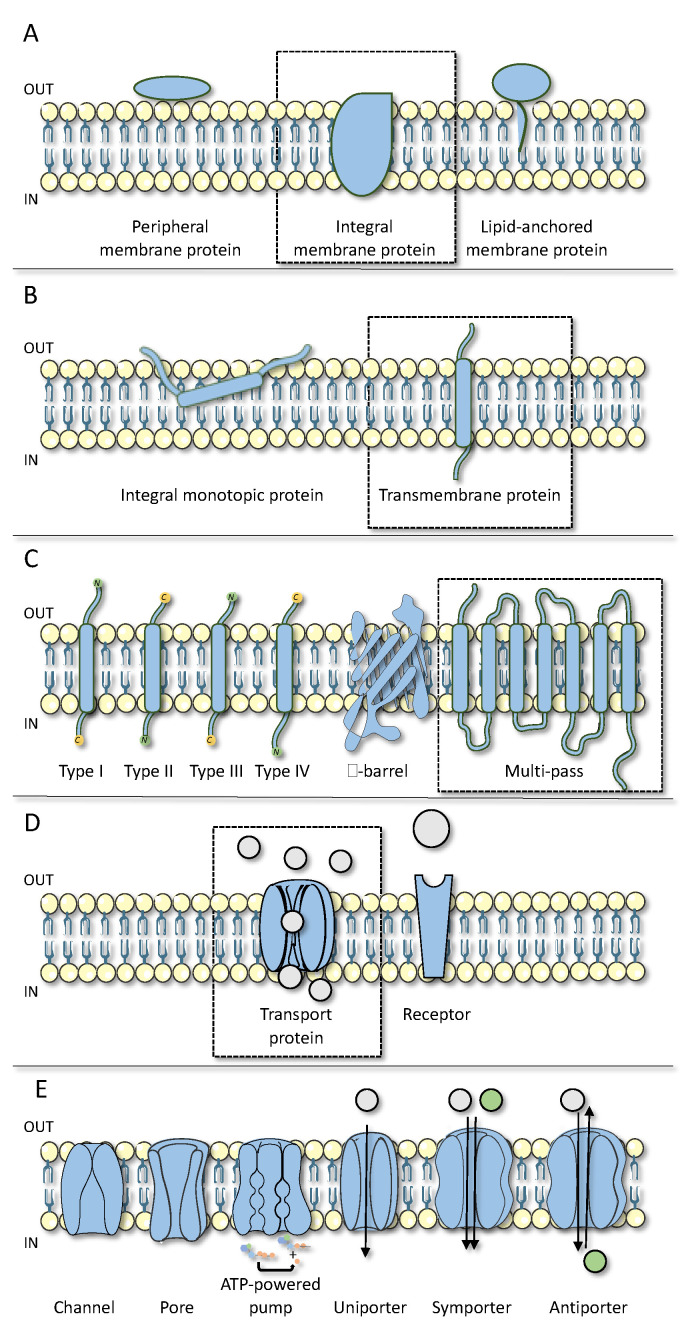
Classification of cell plasma membrane proteins. (**A**) Cell plasma membrane proteins can be divided in 3 general classes, namely integral membrane proteins, peripheral membrane proteins and lipid anchored proteins. (**B**) Integral membrane proteins can be further divided into 2 groups, namely integral monotopic proteins and transmembrane proteins. (**C**) Based on their topology, the family of transmembrane proteins can be classified in type I, type II, type III, type IV, β-barrel and multi-pass transmembrane proteins. (**D**) Multi-pass transmembrane proteins can be divided into different classes based on their function, namely transport proteins and receptors. (**E**) Transport proteins include channels and pores, ATP-powered pumps and 3 types of porters, namely uniporters, symporters and antiporters.

**Figure 2 biomolecules-11-00063-f002:**
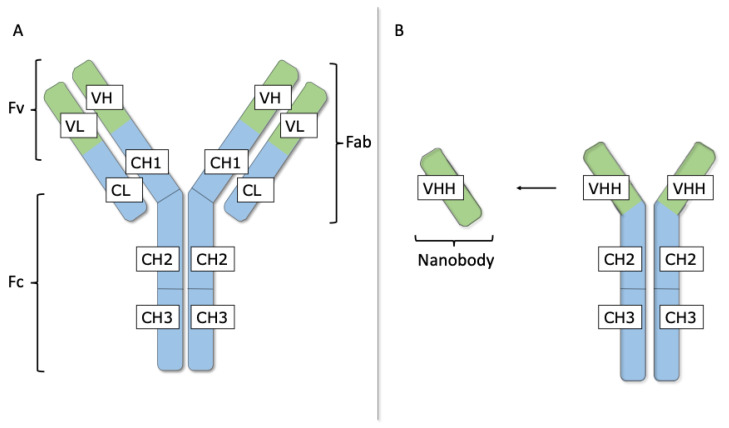
Conventional antibodies and nanobodies. (**A**) Immunoglobulin-G (IgG) with 2 heavy (H) and 2 light (L) chains. The L chain comprises 1 variable (VL) and 1 conserved (CL) domain, whereas the H chain contains 1 variable (VH) and 3 constant (CH1, CH2 and CH3) domains. The paired VH and VL domains form the variable fragment (Fv) and bind to the antigen. The L chain and the first half of the H chain (VH and CH1) are known as the antigen binding fragment (Fab). The CH2 and CH3 domains of the 2 H chains form the crystallizable fragment (Fc). (**B**) Heavy chain-only antibodies (HCAbs) are smaller than conventional antibodies. They are devoid of L chains and the H chain lacks a CH1 domain. HCAbs recognize antigens via the variable domain of the H chain of HCAbs (VHH), also known as a nanobody.

**Table 1 biomolecules-11-00063-t001:** Overview of immunization and biopanning strategies for the identification of cell plasma membrane-protein targeting nanobodies.

Immunization and Biopanning Strategies	Advantages	Disadvantages	References
Transfected cells	- Presents membrane protein in its native form - Applicable for both immunization and panning procedures	- Generation of nanobodies directed towards undesired membrane proteins by the host animal - Requires the construction of transfected cells expressing high levels of the protein of interest	[17,71,84,85,86,87,88]
Membrane extracts	- Presents membrane protein in its native form - Enables to identify intracellular and extracellular binders - Applicable for both immunization and panning procedures	- Generation of nanobodies directed towards undesired membrane proteins by the host animal - Requires the construction of transfected cells expressing high levels of the protein of interest	[71,89,90]
Nanodiscs	- Presents membrane protein in its native form - Enables to identify intracellular and extracellular binders - Does not imply the risk of retrieving unspecific nanobodies that target other components present on cells	- Might require the construction of transfected cells expressing high levels of the protein of interest - Complex membrane model (antigen-dependent)	[91,92,93,94,95,96]
Virus-like particles	- Presents membrane protein in its native form - Applicable for both immunization and panning procedures - Does not imply the risk of retrieving unspecific nanobodies that target other components present on cells	- Complex membrane model (antigen-dependent)	[71,97,98,99]
cDNA immunization	- Presents membrane protein in its native form (in vivo) - No need to construct specific membrane models - No generation of nanobodies directed towards undesired membrane proteins by the host animal	- Difficult retrieval of desired nanobodies due to a weak immune response of the host animal	[68,84,100,101]
Detergent solubilized proteins	- Applicable for both immunization and panning procedures - No need to construct specific membrane models	- Detergents may cause an altered structure and function of the cell plasma membrane proteins	[92]
Endogenous proteins at the cell surface	- Presents membrane protein in its native form - Applicable for both immunization and panning procedures - No need to construct specific membrane models	- Difficult retrieval of desired nanobodies due to a weak immune response of the host animal - Generation of nanobodies directed towards undesired membrane proteins by the host animal - Low expression levels of the antigen of interest	[71,100]

**Table 2 biomolecules-11-00063-t002:** Overview of therapeutic nanobodies targeting cell plasma membrane transport proteins.

Cell Plasma Membrane Transport Proteins	Target	Nanobody Clone	Immunization	Biopanning Strategy	Pharmacological Activity	Reference
Channels and pores	Mouse P2X7	13A7, bivalent 13A7, half-life extended 13A7	P2X7 transfected HEK cells, cDNA + cell boost	Mouse P2X7 transfected HEK cells	Antagonist	[68]
		14D5, bivalent 14D5	P2X7 transfected HEK cells, cDNA + cell boost	Mouse P2X7 transfected HEK cells	Antagonist	[68]
	Human P2X7	Dano1, Dano1-Fc fusion	P2X7 transfected HEK cells, cDNA + cell boost	Human P2X7 transfected HEK cells	Antagonist	[68,102]
	Mouse and human P2X4	Nb271-Fc fusion, Nb284-Fc fusion	cDNA + P2X4 transfected HEK cells	Mouse and human P2X4 transfected HEK cells	Functional effects have not been investigated	[84]
	Human Kv1.3 channels	Monovalent, bivalent, trivalent, half-life and non-half-life extended bivalent nanobodies	Unknown	Unknown	Antagonist	[110]
ATP-powered pumps	Zinc-transporting ATPase ZntA (*Shigella sonnei*) (SsZntA)	Nb9	Purified SsZntA (membrane extracts + detergent)	Biotinylated SsZntA	Antagonist	[114]
	BtuCD-F transporter complex (*Escherichia coli*)	Nb9	Vitamin B12-bound BtuF (detergent)	Vitamin B12-bound BtuF and apo-BtuF	Antagonist	[116]
Porters	Rat vesicular glutamate 1 (VGLUT1)	Nb3, Nb9	Truncated and non-glycosylated mutant of the rat VGLUT1 protein (detergent)	Rat VGLUT1 mutant protein	Antagonist	[121]

## Data Availability

Not Applicable.

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
