# Peer review of "Therapeutic Nanobodies Targeting Cell Plasma Membrane Transport Proteins: A High-Risk/High-Gain Endeavor"

_biomolecules, 2021, doi:10.3390/biom11010063_

Round 1

Reviewer 1 Report

The review article ‘Therapeutic nanobodies targeting cell plasma membrane transport proteins: A high risk / high gain endeavour’ by Raf Van Campenhout et al. provides a comprehensive overview of the nature of plasma membrane proteins, strategies to raise nanobodies against them, and examples of nanobodies against plasma membrane proteins.

In a well-written (and thorough) introduction, membrane-associated and -integrated proteins are described and classified. In a second part, nanobodies are introduced and strategies to raise nanobodies against transmembrane proteins are summarized. In a third part, examples of nanobodies against transmembrane transporters proteins are described in some more detail. A final conclusion evaluates the possibilities to alter the function of membrane transporters, to use nanobodies for targeting, and to modify nanobodies for clinical applications.

A clear asset of this manuscript is the complete overview of existing nanobodies against membrane transport proteins. Style, language and scientific accuracy are at high level. I would only recommend some minor changes, mostly in the general part on antibodies and nanobodies (see below).

The review could be even more helpful to a general readership in two ways:

1) Throughout the manuscript, a number of studies describing nanobodies that interfere with function of membrane transport proteins are mentioned. A more detailed table summarizing the strategy for immunization/library generation/selection, the functional consequences of nanobody binding, and the respective references for each of the major studies would be great resources for readers who are interested in nanobodies against transporters.

2) While the existing nanobodies against transporters are described in detail, the review could dare to make a few more general statements with regards to

  • Mechanisms of how nanobodies can perturb protein function (and how modulatory nanobodies could be identified) 
  • A clear description of scenarios in which nanobodies would be advantageous over conventional antibodies on the one side, and small molecules on the other side. In other words (referring to the title): When is it worth to take the risk to develop nanobodies against membrane transporter? What is the risk and what is the high gain? Which unmet needs could be met with nanobodies?

Specific comments (minor changes, mostly for accuracy/completeness):

  1. Figure 1C/text: The text introduces type I, II, III, and IV single pass proteins, but does not really define them (and the illustration does not explain the difference between type I and III, or II and IV)
  2. Line 159-171: Please adjust the text in this section to avoid a few ambiguities:
  • General: Probably easier to start with a description of the general structure of an antibody and then specify the involved domains / proteolytic fragments (this paragraph is a little convoluted)
  • Line 159: A Fab is not the ‘bifurcated’ end of an IgG (this description would perhaps be correct for a (Fab)2 fragment after Pepsin cleavage).
  • Line 163: a polypeptide chain/loop forms the CDR, but not ‘amino acid sequences’
  1. Line 173: ‘As antibodies can be designed against any possible molecule’. I think we are not advanced enough yet to rationally design antibodies:  The immune response itself as well as the biopanning process select antibodies binding to cognate antigens from the available diversity of antibodies encoded in B cells
  2. Figure legend 2: can be shorter – no need to repeat the same text as in the main text (or be more concise there)
  3. Line 206: While the VHH fragment or nanobody can be described as a single domain antibody, these words are not synonymous, as not every single domain antibody is a nanobody or VHH (Darpins, monobodies etc. are also single domain antibodies). Please be more specific here.
  4. Line 213: If it is still the current conclusion in the field that additional ‘recombination signal sequences’ contribute to nanobody diversity, please cite a primary research article (Nguyen et al.?), rather than a review that cites another review on this matter.
  5. Line 228: Please replace the word ‘production’ with ‘identification’ or ‘generation’, as the production of the protein is not limited (but rather the identification of suitable candidates)
  6. Line 228: There are only few well-described synthetic libraries; probably worth to cite them here
  7. Line 247: omit the word ‘analysis’; it is not the analysis, but the resulting PCR product that is used for library generation
  8. Line 248: bacteria are transformed with nucleic acids (not: nucleic acids are transformed into bacteria
  9. Line 252: Golden gate cloning is a very specific recommendation for a general problem – probably not the scope of this manuscript to be too specific on such technical details (NB: if it matters: we rarely have any clones without insert using conventional cloning)
  10. Line 264: I am not sure one can make the general statement that multi-pass transmembrane never have a folded ectodomain.
  11. Table 1:
  • Include detergent-solubilized proteins as one strategy (the ‘native environment’ is more or less true for all currently described strategies and mostly advantageous when compared to detergent-solubilized protein); using endogenous proteins at the cell surface for selection may also be an option (if completeness is desired)
  • Define ‘membrane extracts’ a little better (the text suggests that this refers to membrane vesicles/fragments from disrupted cells; it could also refer to proteins extracted by membrane solubilization with detergents)
  • Lipid nanodiscs can also be used to embed detergent-enriched proteins – be more specific in the classification
  • Provide references for the individual strategies (this would be a helpful overview for the reader who would like to start generating nanobodies against transmembrane proteins)
  1. Paragraph 4.4.4: Please be more specific in the description of the example(s) of virus-like particles used for immunizations
  2. Paragraph 4.5: In addition to the detailed summary of existing studies, please also make some general statements as to how therapeutic nanobodies may act (e.g. blocking access to binding sites, preventing conformational changes etc.)
  3. Line 473: ‘Besides acting as modulatory therapeutics, nanobodies can be used as tools for targeted therapy.’ Use clearer language to express that the nanobodies can either be used to perturb protein function, or to target some effector molecule to the antigen (I guess the sentence tried to express this, but it is a little subtle).

Reviewer 2 Report

In this review, the authors classified and described cell membrane transport proteins as a therapeutic target via nanobodies. The paper is well written and interesting and provides important and useful information to the readers. However, there are some points that need to be clarified.

  1. Nanobody has a unique feature as 3 CDR in the VHH domain. Moreover, it might have larger and variable CDRs in order to compensate for the lack of VH-VL combinational diversity in IgG. Small-size and convex paratope might allow it to target cryptic epitope in the cell membrane transport proteins, unlike IgG. However, other platform biotechnology such as a cyclic peptide, DNA/RNA aptamer, or chemically programmed antibody would be applicable to recognize the cryptic structure as well as nanobody. Please explain the advantage of the nanobody strategy for targeting membrane transport proteins compared to others described above.
  2. The authors highlighted the active molecular targeting of a nanobody. However, unlike research reagent or extracorporeal diagnostic agent use, the information of pharmacokinetic and biodistribution properties are indispensable for therapeutic use in the human body. In addition, safety, including immunogenicity, is also important, e.g., even CDR in some human antibodies can induce an anti-antibody.
  3. Figure 2 shows a structured comparison of a nanobody and IgG. Readers would be grateful if authors outline the difference in functional features between them.
